# Galectin-9 Induces Mitochondria-Mediated Apoptosis of Esophageal Cancer In Vitro and In Vivo in a Xenograft Mouse Model

**DOI:** 10.3390/ijms20112634

**Published:** 2019-05-29

**Authors:** Taiga Chiyo, Koji Fujita, Hisakazu Iwama, Shintaro Fujihara, Tomoko Tadokoro, Kyoko Ohura, Takanori Matsui, Yasuhiro Goda, Nobuya Kobayashi, Noriko Nishiyama, Tatsuo Yachida, Asahiro Morishita, Hideki Kobara, Hirohito Mori, Toshiro Niki, Mitsuomi Hirashima, Takashi Himoto, Tsutomu Masaki

**Affiliations:** 1Department of Gastroenterology and Neurology, Kagawa University, 1750-1 Ikenobe, Miki-cho, Kita-gun, Kagawa 761-0793, Japan; t_chiyo@med.kagawa-u.ac.jp (T.C.); 92m7v9@med.kagawa-u.ac.jp (K.F.); joshin@med.kagawa-u.ac.jp (S.F.); t-nishioka@med.kagawa-u.ac.jp (T.T.); kyoko_oura@med.kagawa-u.ac.jp (K.O.); s16d729@stu.kagawa-u.ac.jp (T.M.); goda0717@med.kagawa-u.ac.jp (Y.G.); nobuyak@med.kagawa-u.ac.jp (N.K.); n-nori@med.kagawa-u.ac.jp (N.N.); tyachida@med.kagawa-u.ac.jp (T.Y.); asahiro@med.kagawa-u.ac.jp (A.M.); kobara@med.kagawa-u.ac.jp (H.K.); hiro4884@med.kagawa-u.ac.jp (H.M.); mitsuomi@kms.ac.jp (M.H.); 2Life Science Research Center, Kagawa University, 1750-1 Ikenobe, Miki-cho, Kita-gun, Kagawa 761-0793, Japan; iwama@med.kagawa-u.ac.jp; 3Department of Immunology and Immunopathology, Faculty of Medicine, Kagawa University, 1750-1 Ikenobe, Miki-cho, Kita-gun, Kagawa 761-0793, Japan; niki@med.kagawa-u.ac.jp; 4Department of Medical Technology, Kagawa Prefectural University of Health Sciences, 281-1, Hara, Mure-Cho, Takamatsu, Kagawa 761-0123, Japan; himoto@chs.pref.kagawa.jp

**Keywords:** galectin-9, apoptosis, esophageal squamous cell carcinoma, JNK, microRNA

## Abstract

Galectin-9 (Gal-9) enhances tumor immunity mediated by T cells, macrophages, and dendritic cells. Its expression level in various cancers correlates with prognosis. Furthermore, Gal-9 directly induces apoptosis in various cancers; however, its mechanism of action and bioactivity has not been clarified. We evaluated Gal-9 antitumor effect against esophageal squamous cell carcinoma (ESCC) to analyze the dynamics of apoptosis-related molecules, elucidate its mechanism of action, and identify relevant changes in miRNA expressions. KYSE-150 and KYSE-180 cells were treated with Gal-9 and their proliferation was evaluated. Gal-9 inhibited cell proliferation in a concentration-dependent manner. The xenograft mouse model established with KYSE-150 cells was administered with Gal-9 and significant suppression in the tumor growth observed. Gal-9 treatment of KYSE-150 cells increased the number of Annexin V-positive cells, activation of caspase-3, and collapse of mitochondrial potential, indicating apoptosis induction. c-Jun NH_2_-terminal kinase (JNK) and p38 mitogen-activated protein kinase (p38) phosphorylation were activated and could be involved in apoptosis. Therefore, Gal-9 induces mitochondria-mediated apoptosis of ESCC and inhibits cell proliferation in vitro and in vivo with JNK and p38 activation.

## 1. Introduction

Among the cancers of the digestive system, esophageal cancer has poor prognosis and is the sixth most frequent cause of cancer deaths [1], with an estimated 400,000 deaths reported in 2012 worldwide [2].

The standard treatment for locally advanced esophageal squamous cell carcinoma (ESCC) in patients subjected to resection involves preoperative chemotherapy and radiation therapy [3]. In addition, patients that fail to withstand surgery or cannot undergo resection are treated with definitive chemoradiotherapy [4]. Therefore, chemotherapy plays a major role in the treatment of this condition, and the development of new anticancer drugs is important. Recently, mitochondria have attracted attention as a new anticancer, therapeutic target. Mitochondria of cancer cells have altered biological characteristics, including hampered OXPHOS (mitochondrial oxidative phosphorylation system), enhanced rates of glycolysis, hyperpolarization, and changes in the expression of components of the permeability transition pore complex [5]. Even though there are currently no treatments targeting mitochondria, these organelles have the potential to be promising therapeutic targets in the future.

Galectins are proteins that recognize and specifically bind to β-galactosides and function as intercellular signaling factors. Galectin-9 (Gal-9) exhibits several biological activities and plays an important role in eosinophil migration, differentiation, aggregation, adhesion, and apoptosis [6]. Gal-9 was shown to induce apoptosis in Th1 and Th17 cells via the Tim-3 pathway [7,8] and is thought to contribute to the activation of tumor immunity and suppression of the autoimmune response via 4-1BB (CD137) [9]. In addition, Gal-9 has been reported to enhance tumor immunity mediated by T cells, macrophages, and dendritic cells [10] and its expression level in various cancers is correlated with prognosis [11], suggestive of its important role in tumor immunity.

We have recently discovered that Gal-9 induces apoptosis through its direct effect against various cancers, including hepatocellular carcinoma [12,13,14,15]. We performed cluster analysis and functional analysis of microRNAs (miRNAs) relevant to the antitumor effect and reported, for the first time, the possibility of an miRNA-mediated antitumor effect of Gal-9. miRNAs are small endogenous non-coding RNA molecules with a length of 21–30 nucleotides and known to regulate the expression of a variety of target genes at the post-transcriptional and translational levels [16,17]. However, no previous studies have examined miRNAs relevant to the antitumor effect of Gal-9 in ESCC.

The protein Gal-9 exhibits potential applications as a new anticancer agent that activates tumor immunity by directly acting on tumors and exerting its antitumor effects. However, the underlying mechanisms of action and bioactivity of Gal-9 are questionable. No study has investigated the efficacy of Gal-9 against squamous cell carcinomas. In this direction, we aim to analyze the antitumor effect of Gal-9 on ESCC in vitro and in vivo to evaluate the dynamics of apoptosis-related molecules. In addition, we elucidate the mechanism of action of Gal-9 to identify relevant changes in the miRNA expression.

## 2. Results

### 2.1. Galectin-9 (Gal-9) Inhibits the Proliferation of Human Esophageal Squamous Cell Carcinoma (ESCC) Cells In Vitro

We examined the growth inhibitory effect of Gal-9 on ESCC cells with the cell proliferation assay and found that Gal-9 inhibited the proliferation of human ESCC cell lines KYSE-150 and KYSE-180 in a dose- and time-dependent manner (Figure 1A). The effect of Gal-9 was antagonized by 30 mM lactose, indicative of the importance of the β-galactoside–binding properties of Gal-9 for its activity (Figure 1B).

### 2.2. Gal-9 Induces Apoptosis of ESCC Cells

To determine whether Gal-9 induces apoptosis, KYSE-150 cells were incubated with or without 100 nM Gal-9 for 1, 6, 12, and 48 h. Annexin V-Fluorescein isothiocyanate/Propidium Iodide (FITC/PI) staining analysis was performed by flow cytometry. As shown in Figure 2A, the lower left quadrant represents viable cells, the lower right quadrant shows early apoptotic cells, and the upper right quadrant indicates late apoptotic or necrotic cells. The number of cells positive for Annexin V increased the most after 6 h (Figure 2B). The colorimetric assay confirmed the activation of caspase 3 in 6 h, and induction of apoptosis. We failed to detect any increase in the activity of caspase-4, -7, -8 and -9 (Figure 2C).

### 2.3. Gal-9 Changes the Level of Apoptosis-Related Protein

Consistent with the results of the colorimetric assay, Western blot analysis confirmed the increase in the level of cleaved caspase-3 in 6 h (Figure 3A). No difference was observed in the levels of caspase-3, -7, and -9 as well as cleaved caspase-9 and poly ADP ribose polymerase (PARP). We failed to detect cleaved caspase-7 and cleaved-PARP.

We examined the activation of c-Jun NH_2_-terminal kinase (JNK) and p38 mitogen-activated protein kinase (p38), pathways that play a key role in the regulation of apoptosis, by Western blot analysis. No difference was observed in the level of JNK and p38. The expression of phospho- JNK (p-JNK) increased at 6 h, while that of phospho-p38 (p-p38) increased at 6 and 24 h (Figure 3B).

We evaluated the cell fraction for the activation of the intrinsic pathway and release of cytochrome c, Smac/Diablo (Smac), and HtrA2/omi (HtrA2) from the mitochondria to cytoplasm. The membrane fraction contains mitochondria. We found that the levels of cytochrome c, Smac, and HtrA2 were increased in the cytoplasmic fraction at 24 h. (Figure 3C). However, no change in the amount of these proteins in the membrane fraction was detected.

### 2.4. Gal-9 Induces the Collapse of Mitochondrial Potential

Gal-9 can induce the collapse of mitochondrial potential in KYSE-150 cells. In healthy cells with the high membrane potential, JC-1 dye aggregates in the mitochondrial matrix and exhibits an intense red color when viewed with a filter set of 540/570 nm for excitation/emission (Figure 4). In apoptotic cells with the decreased membrane potential, JC-1 dye remains as monomers and exhibits green color using a filter set excitation/emission of 485/535 nm, and therefore green fluorescence was not observed in healthy cells. Gal-9-treated cells showed green staining at 3 h after treatment, which suggests that Gal-9 induced collapse of the mitochondrial potential, and promoted the mitochondrial apoptotic pathway in KYSE-150 cells. 

### 2.5. Effects of Gal-9 on the miRNA Expression

To identify miRNAs related to the antitumor effect of Gal-9, we analyzed miRNA changes following treatment of KYSE-150 cells with Gal-9 using a miRNA array. A significant increase in the expression of miR-222-5p, miR582-5p, miR-6131, and miR-4639-5p was observed (Table 1, GEO, accession no. GSE106842). Unsupervised hierarchical clustering analysis showed that Gal-9–treated group clustered separately from the control group (Figure 5).

### 2.6. Gal-9 Inhibits the Growth of ESCC Cells in the Xenograft Mouse Model

We evaluated if Gal-9 exhibits tumor suppressive activity in vivo. KYSE-150 cells were subcutaneously transplanted into nude mice. After 4 days, the tumor reached a recognizable size of 3 mm. Mice were treated with an intraperitoneal injection of Gal-9. Significant inhibition of the tumor growth was observed in mice treated with Gal-9 as compared with the control mice (Figure 6A,B). All mice survived during the complete study period. Moreover, no sufferance, behavioral alterations, or body weight loss (Figure 6C) was observed in mice treated with Gal-9 as compared with the control mice.

## 3. Discussion

We have previously reported the antitumor activity of Gal-9 in various cancers, including esophageal adenocarcinoma (EAC) [12,13,14,15]. However, studies highlighting the direct effect of Gal-9 on tumor cells are limited to melanoma [18], myeloma [19], and myelogenous leukemia [20]. Therefore, no study has reported the efficacy of Gal-9 against squamous cell carcinomas. The two histological subtypes of esophageal cancer, EAC and ESCC, have different underlying causes, risk factors, and clinicopathological features. In addition, a recent comprehensive analysis using next-generation sequencing has reported the significant differences in genomic changes between EAC and ESCC [21]. Chemotherapy targeting HER-2 (trastuzumab) [22] and vascular endothelial growth factor (ramucirumab) [23,24] is believed to be applicable only to EAC, thereby necessitating separate efficacy studies for ESCC. Although the incidence of EAC has been rising in Western countries [25], ESCC is considered to be the primary histological type in Asian countries, including China and Japan, where it continues to pose serious threats [26]. Therefore, the development of new treatment strategies is desired.

Gal-9, a tandem-repeat-type member of the galectin family, comprises two carbohydrate recognition domains connected by a linker peptide [27]. It was first identified as an eosinophil chemoattractant [28] and is known to induce apoptosis of Th1 and Th17 cells via the Tim-3 pathway [7,8]. Furthermore, it contributes to the activation of tumor immunity and suppression of the autoimmune response via 4-1BB (CD137) [9], indicative of its importance as an immunoregulatory factor. Moreover, Gal-9 is known to improve the pathological condition in animal models of rheumatoid arthritis [8,29] and asthma [30] and may be used for the treatment of autoimmune diseases.

Gal-9 has been reported to enhance the tumor immunity by activating T cells, macrophages, and dendritic cells [10] and, hence, thought to play a key role in tumor immunity. Although no studies have investigated the expression of Gal-9 in ESCC from human subjects, studies of squamous cell carcinomas in other organs have indicated its elevated expression in oral squamous cell carcinoma [31,32] and nasosinusal carcinoma [33] as compared with non-tumor areas. In the cervix, Gal-9 expression is significantly lower in high-grade squamous intraepithelial lesions as compared with low-grade lesions. Similar lower expression was observed in low-grade squamous cell carcinoma than the high-grade squamous cell carcinoma [34]. In general, Gal-9 expression frequently shows a negative correlation with cancer stage, grade, and lymph node metastasis, while high Gal-9 expression is considered to be an indicator of better prognosis [11]. Therefore, it may be suggested that the increase in the expression of Gal-9 in the cancer tissue may be a type of biological defense response.

Gal-9 is believed to induce cancer cell death via an apoptosis signal transduction pathway, thereby suppressing the proliferation of various cancerous cells [12,13,14,15,18,19,20]. In the present study, the proliferation of ESCC cells (KYSE-150 and KYSE-180) was examined following Gal-9 treatment. Cell proliferation was suppressed in a concentration-dependent manner (Figure 1A). At the same time, the effect of Gal-9 was antagonized by 30 mM lactose and was shown to be dependent on β-galactoside binding properties (Figure 1B). Therefore, this was not a nonspecific cytotoxic effect by a random protein. In addition, the tumor progression was significantly suppressed following Gal-9 treatment in the xenograft mouse model established using KYSE-150 cells (Figure 6). Caspase-3 activation was detected with colorimetry and Western blot analysis in KYSE-150 cells treated with Gal-9 (Figure 2C and Figure 3A); an increase in the cells positive for Annexin V was observed at 6 h (Figure 2A,B), indicative of the induction of apoptosis. However, the activation of caspase-4, -7, -8, and -9 was undetected using these methods. Previously, the activation of caspase-3 and -9 was reported in EAC [12], while that of caspase-3, -8, and -9 was observed in myeloma. Apoptosis was inhibited by pan-caspase and caspase-8 inhibitors [19]. However, no inhibition of apoptosis was observed by the pan-caspase inhibitor in melanoma [18], suggesting that the degree, timing, and dependence of caspase activation vary depending on the cancer type or cell line.

The cytoplasmic compartment of KYSE-150 cells showed an elevated expression of cytochrome c, Smac, and HtrA2 after Gal-9 administration (Figure 3C), but no changes in the level of these proteins were detected in the membrane fraction; it is considered that the amount of efflux to the cytoplasm was relatively small compared to that in the membrane fraction. Smac and HtrA2 are mitochondrial proteins that are released into the cytosol together with cytochrome c. These proteins suppress the expression of the inhibitor of apoptosis protein (IAP) and increase caspase activity [35,36,37]. We also discovered that Gal-9 induced collapse of mitochondrial potential in KYSE-150 cells (Figure 4). Since mitochondrial dysregulation is regarded as an irreversible step in apoptosis, it is clear that the treatment of Gal-9 resulted in the activation of mitochondria-mediated apoptosis.

Studies have reported the apoptosis mediated by Tim-3, a physiological ligand of Gal-9 [7,8]; however, we detected no expression of Tim-3 in KYSE-150 cells by real-time PCR, suggesting the involvement of a different signal transduction pathway. In addition, Gal-9 has been reported to induce the activation of JNK and p38 in myeloma, thereby resulting in apoptosis [19]. Here, we found an increase in the level of phosphorylated JNK and p38 in KYSE-150 cells in response to Gal-9 treatment. In general, the activated JNK promotes the release of cytochrome c, Smac, and HtrA2 from the mitochondria via Bid-Bax–dependent mechanism. Furthermore, the phosphorylation of Bad and Bim mediated by JNK and p38 MAPK results in the inhibition of B-cell lymphoma 2 (Bcl-2). In addition, JNK itself phosphorylates Bcl-2 and inhibits its anti-apoptotic activity. The activated JNK is known to translocate into the nucleus and activate transcription factors. JNK may promote apoptosis by increasing the expression of pro-apoptotic genes [38,39,40] (Figure 7). Activation of caspase-9 was not detected in this experimental system. It is conceivable that the peak of activation exists at other time points or that the caspase-9 independent pathway via Smac, HtrA2, an inhibitor of apoptosis protein (IAP) is dominant. Our findings suggest that these changes may result in the activation of the intrinsic pathway, thereby inducing mitochondria-mediated apoptosis. This is the first study to reveal the mechanism underlying Gal-9–mediated apoptosis in solid tumors.

To identify the miRNAs associated with the antitumor effect of Gal-9, a miRNA array was used for the analysis of changes in miRNA expression following treatment of KYSE-150 cells with Gal-9. We found a significant increase in the expression of miR-222-5p, miR582-5p, and miR-6131, while the expression of miR-4639-5p was significantly decreased. The cluster analysis results showed the formation of different clusters for the group treated with Gal-9 as compared with the control group. The miRNA miR-582-5p is known to suppress the proliferation of a variety of tumors, including hepatocellular carcinoma [41], colon cancer [42], and bladder cancer [43], and this suppression is thought to be related to the target genes *CDK1*, *AKT3* [41], *Rab27a* [42], and *FOXC1* [44]. We failed to verify any correlation between other miRNAs and tumor function or prognosis. Based on these findings, we suggest that Gal-9 administration may influence the expression of miRNAs and contribute to the suppression of tumor proliferation.

Previous reports on the pharmacokinetics of Gal-9 have revealed a *C*_max_ value of 35 and 61 ng/mL for the subcutaneous and i.p. injection of 10 μg Gal-9, respectively, in mice [8]. In the present study, gal-9 was administered at 90 μg/mouse dose thrice per week, and no adverse effects such as death or body weight reduction were observed. The same levels of Gal-9 have been administered in several animal studies and no reports of significant adverse effects are available [12,13,14,15,21]. The physiological median value for Gal-9 in human serum was undetected, and the average value was as low as 112 pg/mL [45]. Therefore, the concentration that mediates pharmacological effects without any adverse effects is unclear, thereby necessitating future studies.

## 4. Materials and Methods

### 4.1. Chemicals

A mutant form of Gal-9 that lacks the entire linker region and is highly stable against proteolysis was recombinantly produced as previously described [46]. This protein is known to retain its biological activity.

### 4.2. Cell Lines and Reagents

The human ESCC cell lines KYSE-150 and KYSE-180 were purchased from the Japanese Collection of Research Bioresources (Osaka, Japan). KYSE-150 was cultured in Ham’s F12 and KYSE-180, in Ham’s F12 medium + Roswell Park Memorial Institute (RPMI)-1640 medium (1 to 1 mixture) (Gibco, Carlsbad, CA, USA) supplemented with 2% fetal bovine serum (FBS) (533-69545; Wako, Osaka, Japan) and 100 mg/L penicillin-streptomycin (Invitrogen, Tokyo, Japan) in a humidified atmosphere with 5% of CO_2_ at 37 °C. Cell Counting Kit (CCK)-8 was purchased from Dojindo Laboratories (Kumamoto, Japan).

### 4.3. Cell Proliferation Assay

Cell proliferation assays were performed using CCK-8 according to the manufacturer’s instructions. KYSE-150 and KYSE-180 cells (5000 cells/well) were seeded into each well of a 96-well plate in 100 µL of the growth medium. After 24 h, the cells were treated with 0, 30, 100, or 300 nM Gal-9 and cultured for 24–48 h. In addition, 30 mM of lactose was added to inhibit the binding of Gal-9 and sucrose was used as a control [47]. Cells were treated with CCK-8 reagent (10 μL) for 2 h, and the absorbance was measured for each well at 450 nm wavelength using an auto-microplate reader.

### 4.4. Apoptosis Analysis

The state of apoptosis was analyzed by Annexin V-biotin apoptosis detection kit (BioVision, Milpitas, CA, USA) following double staining with fluorescein isothiocyanate (FITC)-conjugated Annexin V and propidium iodide (PI). Annexin V binds to apoptotic cells with exposed phosphatidylserine, while PI binds to the late apoptotic and necrotic cells with membrane damage. ESCC cells were treated for 1, 6, 12, and 48 h. Staining was performed according to the manufacturer’s instructions.

### 4.5. Colorimetric Assay for Caspase-3, -4, -7, -8 and -9 Activity

The activities of caspase-3, -4, -7, -8 and -9 were analyzed using caspase-3, -4, -8, and -9 colorimetric assay kits and a caspase-7 immunoassay kit (BioVision, Milpitas, CA, USA), respectively. KYSE-150 cells were cultured in the presence or absence of 100 nM Gal-9 for 6 h. The subsequent procedures were performed according to the manufacturer’s instructions.

### 4.6. Gel Electrophoresis and Western Blot Analysis

The following antibodies were used: β-actin (Sigma-Aldrich, St. Louis, MO, USA; A5441), caspase-3 (8G10) (#9665), cleaved caspase-3 (D175) (#1661), caspase-7 (D2Q3L) (#12827), cleaved caspase-7 (Asp198) (D6H1) (#8438), caspase-9 (C9) (#9508), cleaved caspase-9 (Asp330) (#7237), poly (ADP-ribose) polymerase (PARP, #9542), cleaved-PARP (D64E10) (#5625), cytochrome c (D18C7) (#11940), Smac (#2954), HtrA2/Omi (HtrA2) (D20A5) (#9745), stress-activated protein kinase/c-Jun NH_2_-terminal kinase (SAPK/JNK, #9252), phospho-SAPK/JNK (Thr183/Tyr185) (81E11) (#4668), p38 mitogen-activated protein kinase (MAPK, D13E1) (#8690), phospho-p38 MAPK (Thr180/Tyr182) (D3F9) (#4511), and secondary horseradish peroxidase (HRP)-linked anti-mouse and anti-rabbit IgG antibodies (Cell Signaling Technology, Danvers, MA, USA). Chemiluminescence reagents for Western blot analysis were purchased from Perkin-Elmer Co. (Waltham, MA, USA).

KYSE-150 cells treated with or without 100 nM Gal-9 were cultured for 24 h. The cells were solubilized in a lysis buffer (PRO-PREP, iNtRON Biotechnology, Seongnam, Korea) and the cell lysate was centrifuged at 13,000× *g* at 4 °C for 5 min. The supernatant obtained was used for Western blot analysis. Cell fractionation was performed according to the manufacturer’s instructions using a cell fractionation kit (#9038, Cell Signaling Technology).

The protein concentration was determined with NanoDrop 2000 Fluorospectrometer (Thermo Scientific Corporation, Waltham, MA, USA). After adding 2× sodium dodecyl sulfate (SDS) sample buffer, the samples were heated to 95–100 °C for 5 min and then cooled on ice. The samples were electrophorized using 10% SDS polyacrylamide gel electrophoresis (SDS-PAGE) and the proteins were transferred onto nitrocellulose membranes. The membranes were incubated with primary antibodies after blocking. Following washes, the blots were reacted with HRP-conjugated secondary antibodies. The protein bands were visualized with an enhanced chemiluminescence detection system (Perkin-Elmer Co., Waltham, MA, USA) on X-ray film. All experiments were repeated thrice.

### 4.7. Mitochondrial Membrane Potential Assay

The collapse of mitochondrial potential induced by galectin-9 treatment was visualized using JC-1 Mitochondrial membrane potential assay kit (Cayman Chemical, Ann Arbor, MI, USA) according to the manufacturer’s protocol. Briefly, KYSE-150 cells (2.0 × 10^5^ cells in a 60-mm dish) were treated with or without 100 nM Gal-9 for 3 h. Afterward, cells were incubated with JC-1 dye for 15 min. Fluorescence of JC-1 monomers was detected using a filter set excitation/emission 485/535 nm. JC-1 aggregates were detected using a filter set excitation/emission 540/570 nm.

### 4.8. Analysis of miRNA Microarray

Total RNA was extracted from KYSE-150 cells (1.0 × 10^6^ cells in a 100-mm dish) treated with or without 100 nM Gal-9 for 6 h using the miRNeasy mini kit (Qiagen, Hilden, Germany) according to the manufacturer’s instructions. RNA amount was quantified with an RNA 6000 Nano kit (Agilent Technologies, Santa Clara, CA, USA) and the samples were labeled using miRCURY Hy3/Hy5 Power labeling kit (Exiqon, Vedbaek, Denmark), followed by the hybridization with a human miRNA Oligo chip (v.21.0; Toray Industries, Tokyo, Japan). The chips were scanned by 3D-Gene Scanner 3000 (Toray Industries). The raw intensity of the image was read using 3D-Gene Extraction Version 1.2 software (Toray Industries). The changes in the miRNA expression between Gal-9–treated and control samples were analyzed with GeneSpring GX v 10.0 (Agilent Technologies). Quantile normalization was performed on the background subtraction data. Differences in miRNA expression were tested by Mann–Whitney *U* test. Hierarchical clustering was performed using the furthest neighbor method with the absolute uncentered Pearson’s correlation coefficient as a metric. A heat map with relative expression intensity of each miRNA was generated, wherein the base-2 logarithm of the intensity was median-centered for each row.

### 4.9. Xenograft Model Analysis

Animal experiments were approved by the Committee on Experimental Animals of Kagawa University (Kagawa, Japan) (13 April 2019 approval code A22). Male athymic mice (BALB/c-nu/nu; 6-weeks old; 20–25 g) were purchased from Japan SLC (Shizuoka, Japan). The mice were provided with free access to sterilized food (gamma ray-irradiated food, CL-2; CLEA Japan Inc., Tokyo, Japan) and autoclaved water. Each mouse was subcutaneously injected with KYSE-150 cells (5 × 10^6^ cells/animal) in the flank. Five days later, the xenografts were observed as a mass of 3 mm (maximum) diameter. The mice were randomly assigned to two groups of seven mice each. Gal-9 treatment group (*n* = 7) were intraperitoneally (i.p.) injected with Gal-9 (90 μg/body) thrice a week. Only phosphate-buffered saline was i.p. administered to the control group (*n* = 7). The tumor size was measured twice a week by measuring the two largest perpendicular dimensions. The tumor volume was calculated as follows: tumor volume (mm^3^) = (tumor length [mm] × square of tumor width [mm]^2^)/2. All mice were sacrificed on day 43 after treatment.

### 4.10. Real-Time Polymerase Chain Reaction (PCR)

Reverse-transcription and real-time quantitative PCR were performed using the ΔΔ*C*_t_ method to evaluate the expression of Tim-3 in KYSE-150 and KYSE-180 cells. Total RNA was extracted using the miRNeasy mini kit (Qiagen, Venlo, The Netherlands). TaqMan^®^ Gene Expression Assays (Life Technologies, Carlsbad, CA, USA) were adapted to determine the expression level of mRNAs of Tim-3 with β-actin as an internal control according to the manufacturer’s protocol. Healthy human peripheral blood mononuclear cells (PBMCs) were used as positive control.

### 4.11. Statistical Analyses

All statistical analyses were performed by GraphPad Prism 7.0 (GraphPad Software, La Jolla, CA, USA). Comparisons between two groups were performed by unpaired Student’s *t*-tests. Two-way analysis of variance (ANOVA) was used to compare the cell viability and tumor volume of the treated and control mice. In all analyses, a value of *p* < 0.05 was considered significant.

## 5. Conclusions

In conclusion, this study clearly showed, for the first time, that Gal-9 induces mitochondrial-mediated apoptosis in esophageal squamous cell carcinoma cells. Furthermore, it was suggested that JNK and p38 are involved in its mechanisms. Our findings showed that Gal-9 might be a promising candidate for anti-ESCC therapy, with a novel mechanism of action via mitochondria.

## Figures and Tables

**Figure 1 ijms-20-02634-f001:**
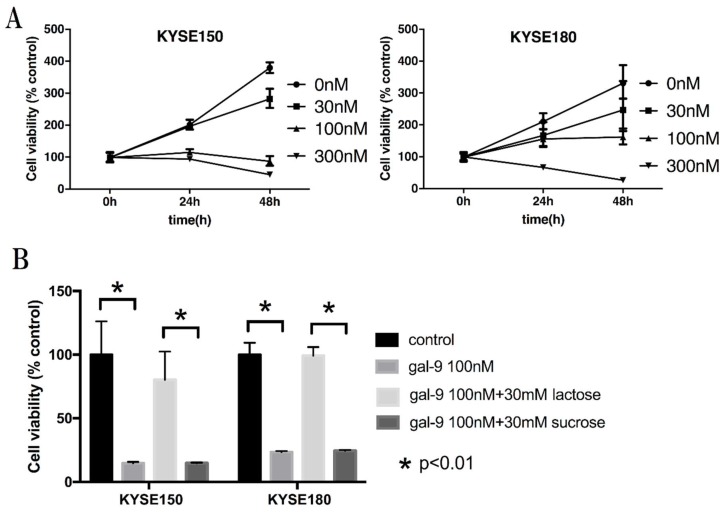
Galectin-9 (Gal-9) suppresses the proliferation of esophageal squamous cell carcinoma (ESCC) cells in vitro. (**A**) Human ESCC cell lines (KYSE-150 and KYSE-180) were treated with Gal-9 (30, 100, and 300 nM) and the cell viability was measured with Cell Counting Kit-8 after 24 and 48 h. The tumor cell proliferation was significantly suppressed in a time- and concentration-dependent manner. The results are expressed as the percentage of viable cells relative to the control cells (0 nM); (**B**) The effect of Gal-9 was antagonized by 30 mM lactose in both KYSE-150 and KYSE-180 cells in 48 h, suggesting that the β-galactoside–binding properties of Gal-9 are essential for its activity. The error bars represent 95% confidence interval (CI).

**Figure 2 ijms-20-02634-f002:**
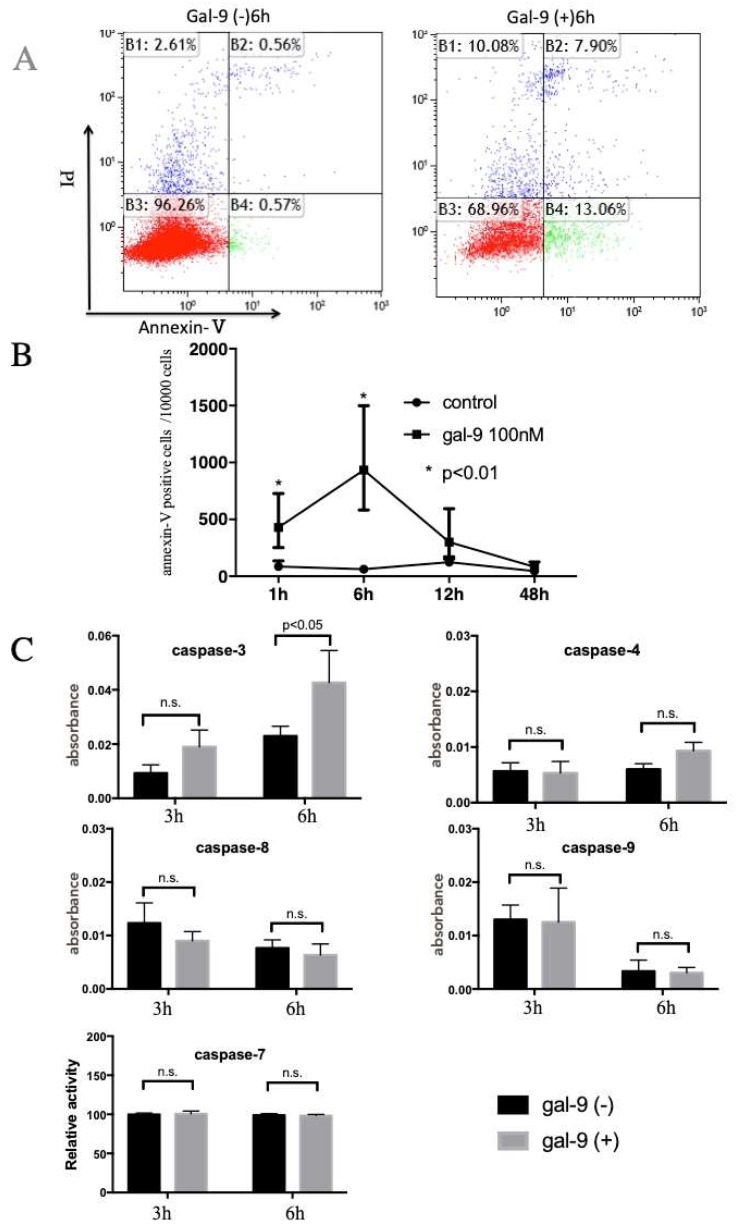
Gal-9 induces apoptosis of ESCC cells. (**A**) Early apoptosis analysis in KYSE-150 cells 6 h after the administration of Gal-9. The lower left quadrant represents viable cells (red dots), the lower right quadrant early apoptotic cells (green dots), and the upper right quadrant indicates the late apoptotic or necrotic cells (blue dots); (**B**) The transition of Annexin V-positive cell number following treatment with Gal-9 for 1, 6, 12, and 48 h. Annexin V-positive cells were significantly increased at 1 h and 6 h after Gal-9 treatment, compared to the control; (**C**) The activation of caspase-3 was observed after 6 h of treatment with 100 nM Gal-9, as evaluated with the colorimetric assay. No significant change in the activity of caspase-4, -7, -8 and -9 was observed. The error bars represent 95% CI.

**Figure 3 ijms-20-02634-f003:**
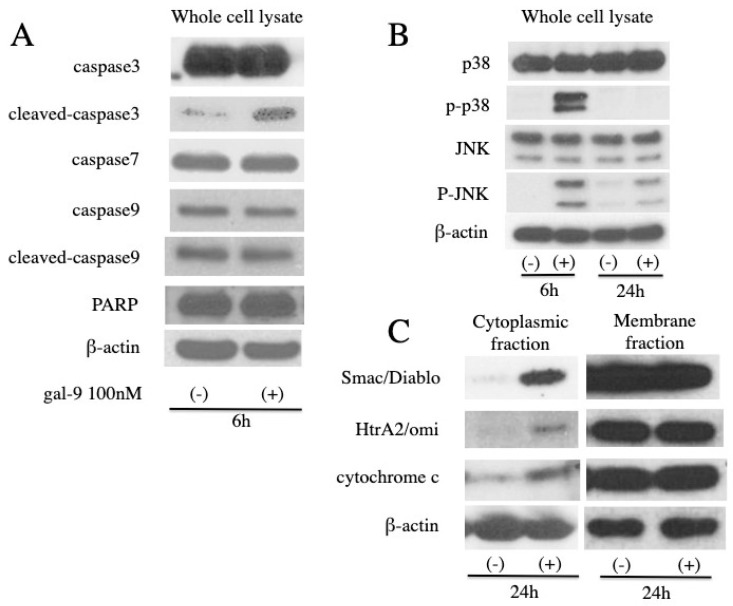
Western blotting analysis of apoptosis-related regulators. The lysate of KYSE-150 cells treated with or without 100 nM Gal-9 was analyzed by Western blotting. (**A**) Cleaved caspase-3 level was increased at 6 h after Gal-9 treatment; (**B**) No difference was observed in the total amount of c-Jun NH_2_-terminal kinase (JNK) and p38. Phospho-SAPK/JNK (p-JNK) increased in 6 h, while phospho-p38 MAPK (p-p38) increased in 6 and 24 h (**B**); (**C**) An increase in Smac, HtrA2, and cytochrome c level was observed in the cytoplasmic fraction, but no change in the levels of these proteins was observed in the membrane fraction containing mitochondria.

**Figure 4 ijms-20-02634-f004:**
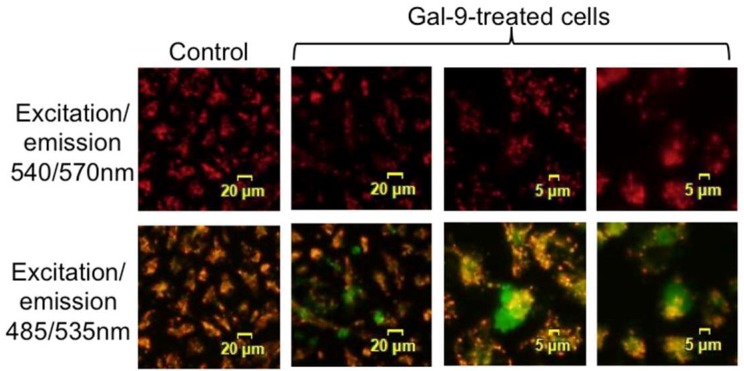
Gal-9 induces the collapse of the mitochondrial potential. KYSE-150 cells were treated with 100 nM Gal-9 for 3 h and then treated with JC-1 dye for 15 min. Cells were then visualized by a fluorescent microscope to examine the conversion of red to green color in healthy and apoptotic cells, respectively. Healthy cells were stained red with JC-1 dye using a filter set excitation/emission 540/570 nm. Apoptotic cells were stained green with JC-1 dye using a filter set excitation/emission 485/535 nm, suggesting that a collapse of mitochondrial potential by Gal-9 treatment. Scale bars are shown for the images.

**Figure 5 ijms-20-02634-f005:**
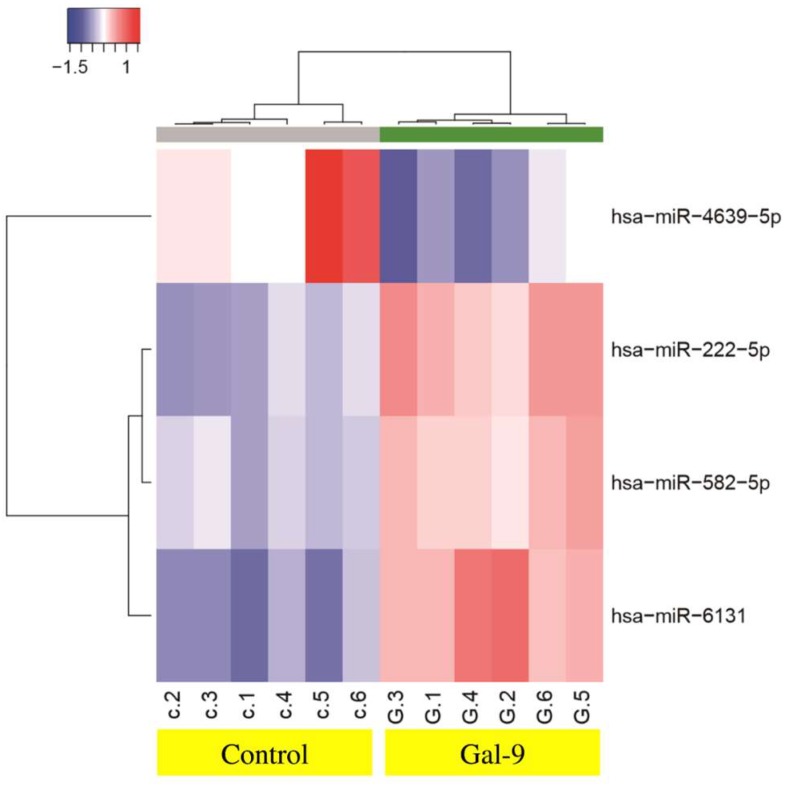
Hierarchical clustering analysis of miRNA expression in KYSE-150 with or without Gal-9 treatment. The miRNA expression clustered according to the expression profiles of four differentially expressed miRNAs between KYSE-150 cells treated with or without Gal-9. The analyzed samples are shown as the columns and the miRNAs are shown as separate rows. The color scale at the top indicates the relative expression levels of the miRNAs; red indicates high expression level and blue, low expression level.

**Figure 6 ijms-20-02634-f006:**
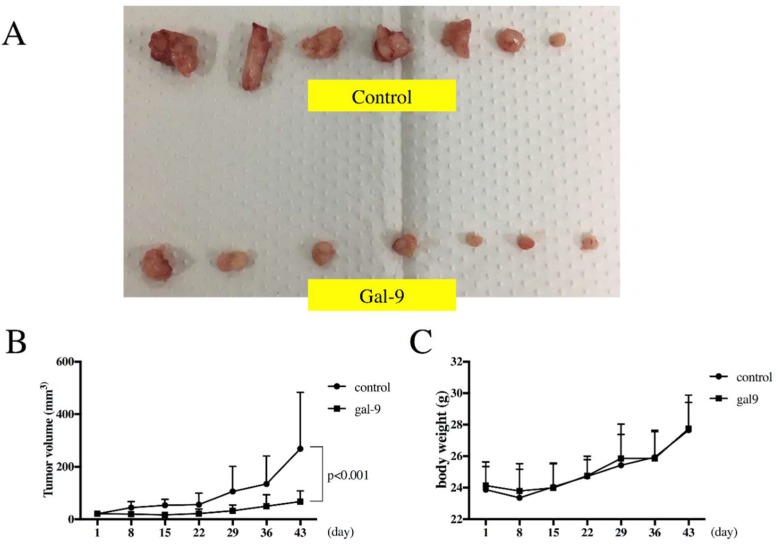
Gal-9 inhibits the growth of ESCC cells in the xenograft mouse model. (**A**) Image of tumors excised from Gal-9–treated and control (only PBS) mice; (**B**) Mean tumor volume of Gal-9 treated mice and control mice. KYSE-150 cells (5 × 10^6^ cells/animal) were subcutaneously injected into the flank of each mouse. Gal-9 was administered on day 5 after the tumor mass was formed. The horizontal axis shows the days of the administration of Gal-9. Tumor growth was significantly suppressed in the Gal-9 treated mice when compared to control mice; (**C**) Changes in the body weight in control and Gal-9–treated mice. No difference was observed in both groups during the experimental period. The error bars represent the standard deviation (SD).

**Figure 7 ijms-20-02634-f007:**
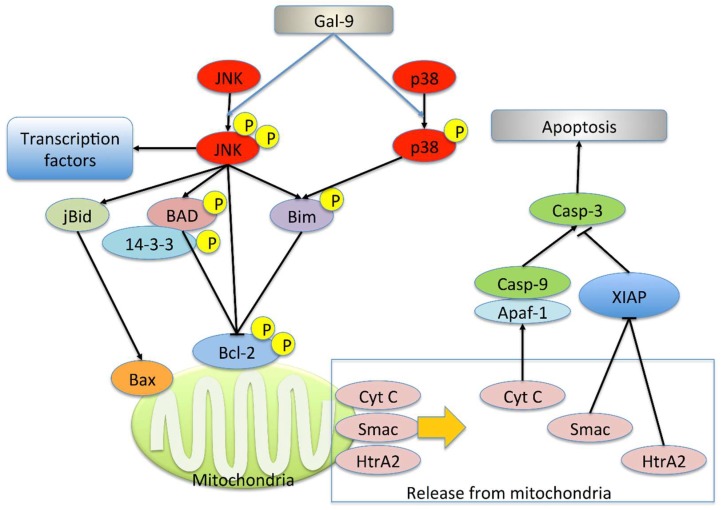
Mitochondria-mediated apoptotic signaling pathway by JNK and p38. Activated JNK promotes the release of cytochrome c (Cyt C), Smac/Diablo (Smac), and HtrA2 from mitochondria via the Bid-Bax–dependent mechanism. Cyt C initiates the activation of the caspase-9–dependent caspase cascade. Smac and HtrA2 inhibit inhibitor of apoptosis protein (IAP) and promotes apoptosis. Furthermore, JNK phosphorylates Bad and its sequestering partner 14–3-3 and promotes Bad-mediated neutralization of Bcl2 family. JNK may also phosphorylate Bcl-2 and inhibit its anti-apoptotic activity. Both JNK and p38 MAPK (p38) inhibit Bcl-2 by phosphorylating Bim. In addition, the activated JNK translocates to the nucleus and activates transcription factors. JNK may promote apoptosis by increasing the expression of pro-apoptotic genes.

**Table 1 ijms-20-02634-t001:** Statistical results of miRNAs in KYSE-150 cells treated with or without Gal-9.

Downregulated	Fold (Treated/Untreated)	*p* Value	Chromosomal Localization
miR-4639-5p	0.45	0.0022	6
**Upregulated**			
miR-222-5p	1.81	0.0022	X chromosome
miR-582-5p	1.52	0.0022	5
miR-6131	2.27	0.0022	5

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
