# Peer review of "Galectin-9 Induces Mitochondria-Mediated Apoptosis of Esophageal Cancer In Vitro and In Vivo in a Xenograft Mouse Model"

_ijms, 2019, doi:10.3390/ijms20112634_

Round 1

Reviewer 1 Report

Comment on Figure 6: if I understand correctly, the authors present excised tumours from the same group after 43 days. There is a significant difference between the volumes of tumors of control mice. In fact, for the last two control tumours, the difference with the treated tumors is not significant in this photo. How do the authors explain this?

Author Response

 Thank you for your kind comments. Following the advice, the experts recalibrated the paper. Mouse xenograft models originally have variations in tumor size. Because there is variation in the graft survival rate. Therefore, in order to enable these comparisons, the planted mice were randomly divided into two groups and a statistically significant difference was detected. Due to the nature of the experimental system, it is inevitable that both groups contain small tumor masses with poor engraftment.

Reviewer 2 Report

Introduction

Overall, the authors have provided strong evidence that Gal- induces apoptosis in esophageal cancer cell lines and in vivo xenograft models of these cancer cell lines. Furthermore, they provided strong evidence that Gal-9 uses intrinsic apoptotic pathway independent of the extrinsic apoptotic pathway. They also showed that both the p38 and JNK pathways maybe involved in the mechanisms of induction of apoptosis by Gal-9. They showed strong evidence of caspase3 activation. In addition, by analyzing mitochondrial proteins released to the cytosol, the showed that treatment of cells with gal-9 induced mitochondrial release of Smac/Diablo, HtrA2/omi and cytochrome to the cytosol, further supporting their conclusion that gal-9 induces mitochondrial-mediated apoptosis of esophageal cancer in vitro and in vivo in a xenograft moue model.   Lastly, they also showed strong data indicating that Gal-9 regulates some members of mi-RNA family in particular mi-222.5p, miR-582-5p and miR-6131.

Specific Strengths:

            Specifically, in the revised version, the authors have addressed the reviewer’s concerns.  They provided additional citations in response to some of the questions raised by the reviewer. Furthermore, the authors followed the recommendations of the reviewer and conducted additional experiments to clarify the roles of mitochondria and JNK pathway in gal-9 induced apoptosis. Related to this, they were successful regarding the Western blot analysis of proteins in the cytoplasmic fractions. Here, they demonstrated clear increases in levels Smac/Diablo, HtrA2/omi and cytochrome c in the cytoplasmic fraction in response to gal-9. However, they could not show that there is associated decline in membrane levels of these proteins. This failure maybe because their Western blot bands in the mitochondrial membrane were definitely overexposed, thus making it impossible to detect small decline in membrane protein levels.    In case, their results from the cytoplasmic protein analysis indicate mitochondrial involvement in the mechanisms of gal-9-induced apoptosis. They indicated having difficulties in conducting the JNK inhibition experiment and proposed to take it up at another time.

Their data presentations are acceptable. The article is well-written, and their conclusions match their evidence. The outcomes from their studies are highly relevant and could contribute interesting information towards advancement of the field. Despite the unfavorable comes from the JNK inhibition experiments, I strongly believe that the strengths of the scientific contributions from this study are significant.

Author Response

We would like to thank you very carefully for your kind review.

Galectin-9 has been shown to increase tumor immunity in vivo in the previous report, and have a direct anti-tumor effect as reported in our present report. These action mechanisms are considered to be different mechanisms from existing anticancer agents, and we believe that combining them with existing antitumor treatments can provide high therapeutic effects. We will investigate further and lead to development of new treatment

Round 2

Reviewer 1 Report

This article is now acceptable for publication in IJMS